# QTRAN++: Improved Value Transformation for Cooperative Multi-Agent Reinforcement Learning

## Abstract

QTRAN is a multi-agent reinforcement learning (MARL) algorithm capable of learning the largest class of joint-action value functions up to date. However, despite its strong theoretical guarantee, it has shown poor empirical performance in complex environments, such as Starcraft Multi-Agent Challenge (SMAC). In this paper, we identify the performance bottleneck of QTRAN and propose a substantially improved version, coined QTRAN++. Our gains come from (i) stabilizing the training objective of QTRAN, (ii) removing the strict role separation between the action-value estimators of QTRAN, and (iii) introducing a multi-head mixing network for value transformation. Through extensive evaluation, we confirm that our diagnosis is correct, and QTRAN++ successfully bridges the gap between empirical performance and theoretical guarantee. In particular, QTRAN++ newly achieves state-of-the-art performance in the SMAC environment. The code will be released.

## 1 Introduction

Over the decade, reinforcement learning (RL) has shown successful results for single-agent tasks (Mnih et al., 2015; Lillicrap et al., 2015). However, the progress in multi-agent reinforcement learning (MARL) has been relatively slow despite its importance in many applications, *e.g.,* controlling robot swarms (Yogeswaran et al., 2013) and autonomous driving (Shalev-Shwartz et al., 2016). Indeed, naïvely applying single-agent algorithms demonstrated underwhelming results (Tan, 1993; Lauer and Riedmiller, 2000; Tampuu et al., 2017). The main challenge is handling the non-stationarity of the policies: a small perturbation in one agent's policy leads to large deviations of another agent's policy.

Centralized training with decentralized execution (CTDE) is a popular paradigm for tackling this issue. Under the paradigm, value-based methods (i) train a central action-value estimator with access to full information of the environment, (ii) decompose the estimator into agent-wise utility functions, and (iii) set the decentralized policy of agents to maximize the corresponding utility functions. Their key idea is to design the action-value estimator as a decentralizable function (Son et al., 2019), i.e., individual policies maximizing the estimate of central action-value. For example, value-decomposition networks (VDN, Sunehag et al. 2018) decomposes the action-value estimator into a summation of utility functions, and QMIX (Rashid et al., 2018) use a monotonic function of utility functions for the decomposition instead. Here, the common challenge addressed by the prior works is how to design the action-value estimator as flexible as possible, while maintaining the execution constraint on decentralizability.

Recently, Son et al. (2019) proposed QTRAN to eliminate the restriction of value-based CTDE methods for the action-value estimator being decentralizable. To be specific, the authors introduced a true action-value estimator and a transformed action-value estimator with inequality constraints imposed between them. They provide theoretical analysis on how the inequality constraints allow QTRAN to represent a larger class of estimators than the existing value-based CTDE methods. However, despite its promise, other recent studies have found that QTRAN performs empirically worse than QMIX in complex MARL environments (Mahajan et al., 2019; Samvelyan et al., 2019; Rashid et al., 2020a). Namely, a gap between the theoretical analysis and the empirical observation is evident, where we are motivated to fill this gap.

**Contribution.** In this paper, we propose QTRAN++, a novel value-based MARL algorithm that resolves the limitations of QTRAN, i.e., filling the gap between theoretical guarantee and empirical performance. Our algorithm maintains the theoretical benefit of QTRAN for representing the largest class of joint action-value estimators, while achieving state-of-the-art performance in the popular complex MARL environment, StarCraft Multi-Agent Challenge (SMAC, Samvelyan et al. 2019).

At a high-level, the proposed QTRAN++ improves over QTRAN based on the following critical modifications: (a) enriching the training signals through the change of loss functions, (b) allowing shared roles between the true and the transformed joint action-value functions, and (c) introducing multi-head mixing networks for joint action-value estimation. To be specific, we achieve (a) through enforcing additional inequality constraints between the transformed action-value estimator and using a non-fixed true action-value estimator. Furthermore, (b) helps to maintain high expressive power for the transformed action-value estimator with representation transferred from the true action-value estimator. Finally, (c) allows an unbiased credit assignment in tasks that are fundamentally non-decentralizable.

We extensively evaluate our QTRAN++ in the SMAC environment, where we compare with 5 MARL baselines under 10 different scenarios. We additionally consider a new rewarding mechanism which promotes "selfish" behavior of agents by penalizing agents based on their self-interest. To highlight, QTRAN++ consistently achieves state-of-the-art performance in all the considered experiments. Even in the newly considered settings, our QTRAN++ successfully trains the agents to cooperate with each other to achieve high performance, while the existing MARL algorithms may fall into local optima of learning selfish behaviors, e.g., QMIX trains the agents to run away from enemies to preserve their health conditions. We also construct additional ablation studies, which reveal how the algorithmic components of QTRAN++ are complementary to each other and crucial for achieving state-of-the-art performance. We believe that QTRAN++ method can be a strong baseline when other researchers pursue the MARL tasks in the future.

## 2 PRELIMINARIES

### 2.1 PROBLEM STATEMENT

In this paper, we consider a decentralized partially observable Markov decision process (Oliehoek et al., 2016) represented by a tuple $\mathcal{G} = \langle \mathcal{S}, \mathcal{U}, P, r, O, N, \gamma \rangle$. To be specific, we let $s \in \mathcal{S}$ denote the true state of the environment. At each time step, an agent $i \in \mathcal{N} := \{1, ..., N\}$ selects an action $u_i \in \mathcal{U}$ as an element of the joint action vector $\boldsymbol{u} := [u_1, \cdots, u_N]$. It then goes through a stochastic transition dynamic described by the probability $P(s'|s, \boldsymbol{u})$. All agents share the same reward $r(s, \boldsymbol{u})$ discounted by a factor of $\gamma$. Each agent $i$ is associated with a partial observation $O(s, i)$ and an action-observation history $\tau_i$. Concatenation of the agent-wise action-observation histories is denoted as the overall action-observation history $\boldsymbol{\tau}$.

We consider value-based policies under the paradigm of centralized training with decentralized execution (CTDE). To this end, we train a joint action-value estimator $Q_{\mathtt{jt}}(s, \boldsymbol{\tau}, \boldsymbol{u})$ with access to the overall action-observation history $\boldsymbol{\tau}$ and the underlying state $s$. Each agent $i$ follows the policy of maximizing an agent-wise utility function $q_i(\tau_i, u_i)$ which can be executed in parallel without access to the state $s$. Finally, we denote the joint action-value estimator $Q_{\mathtt{jt}}$ to be *decentralized* into agent-wise utility functions $q_1, \ldots, q_N$ when the following condition is satisfied:

$$\arg\max_{\boldsymbol{u}} Q_{\mathtt{jt}}(s, \boldsymbol{\tau}, \boldsymbol{u}) = \big[ \arg\max_{u_1} q_1(\tau_1, u_1), \ldots, \arg\max_{u_N} q_N(\tau_N, u_N) \big]. \tag{1}$$

Namely, the joint action-value estimator is decentralizable when there exist agent-wise policies maximizing the estimated action-value.

### 2.2 RELATED WORK

**QMIX.** Rashid et al. (2018) showed how decentralization in Equation 1 is achieved when the joint action-value estimator is restricted as a non-decreasing *monotonic* function of agent-wise utility functions. Based on this result, they proposed to parameterize the joint action-value estimator as a *mixing network* $f_{\mathtt{mix}}$ for utility functions $q_i$ with parameter $\theta$:

$$Q_{\mathtt{jt}}(s, \boldsymbol{\tau}, \boldsymbol{u}) = f_{\mathtt{mix}}\big(q_1(\tau_1, u_1), \ldots, q_N(\tau_N, u_N); \theta(s)\big), \tag{2}$$

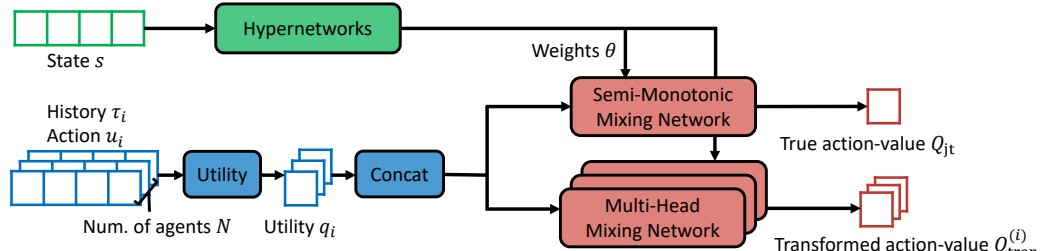

Figure 1: Architecture of QTRAN++

To be specific, QMIX express the mixing network $f_{\texttt{mix}}$ as a fully-connected network with non-negative weight $\theta(s) \geq 0$. The non-negative weight is state-dependent since it is obtained as an output of hypernetworks (Ha et al., 2017) with the underlying state as input.

**QTRAN.** In contrast to QMIX, QTRAN (Son et al., 2019) does not restrict the joint action-value $Q_{\texttt{jt}}$ to a certain class of functions. Instead, QTRAN newly introduces a decentralizable *transformed* action-value estimator $Q_{\texttt{tran}}$ expressed as a linear summation over the utility functions and an additional value estimator:

$$Q_{\texttt{tran}}(\boldsymbol{\tau}, \boldsymbol{u}) = \sum_{i \in \mathcal{N}} q_i(\tau_i, u_i) + V(\boldsymbol{\tau}),$$

where the value estimator $V(\boldsymbol{\tau})$ act as a bias on the action-value estimate independent from actions. Then for each sample $(\boldsymbol{\tau}, \boldsymbol{u})$ collected from a replay buffer, QTRAN trains the transformed action-value estimator to approximate the behavior of "true" action-value estimator $Q_{\texttt{jt}}$ using two loss functions $\mathcal{L}_{\texttt{opt}}$ and $\mathcal{L}_{\texttt{nopt}}$:

$$
\begin{aligned}
\mathcal{L}_{\texttt{opt}}(\boldsymbol{\tau}) &= \left( Q_{\texttt{jt}}(\boldsymbol{\tau}, \bar{\boldsymbol{u}}) - Q_{\texttt{tran}}(\boldsymbol{\tau}, \bar{\boldsymbol{u}}) \right)^2, \\
\mathcal{L}_{\texttt{nopt}}(\boldsymbol{\tau}, \boldsymbol{u}) &= \left( Q_{\texttt{clip}}(\boldsymbol{\tau}, \boldsymbol{u}) - Q_{\texttt{tran}}(\boldsymbol{\tau}, \boldsymbol{u}) \right)^2, \\
Q_{\texttt{clip}}(\boldsymbol{\tau}, \boldsymbol{u}) &= \max\{Q_{\texttt{jt}}(\boldsymbol{\tau}, \boldsymbol{u}), Q_{\texttt{tran}}(\boldsymbol{\tau}, \boldsymbol{u})\},
\end{aligned}
\tag{3}
$$

where $\bar{\boldsymbol{u}}$ is the "optimal" action maximizing the utility functions $q_i(\tau_i, u_i)$ for $i \in \mathcal{N}$. Note that QTRAN does not update the true action-value estimator $Q_{\texttt{jt}}$ using $\mathcal{L}_{\texttt{opt}}$ and $\mathcal{L}_{\texttt{nopt}}$.

**Other CTDE algorithms.** Several other value-based methods have been proposed under the CTDE paradigm. Value decomposition network (VDN, Sunehag et al. 2018) is a predecessor of QMIX which factorize the action-value estimator into a linear summation over the utility functions. Due to this structural constraint, VDN often suffers from low expressive power.

Most recently, QPLEX (Wang et al., 2020a) and Weighted QMIX (Rashid et al., 2020b) appeared as a new framework for removing the restriction of the joint action-value estimator $Q_{\texttt{jt}}$ to a certain class of functions. To this end, QPLEX decomposes $Q_{\texttt{jt}}$ into a sum of a value function and a non-positive advantage function. The main role of non-positive advantage function is to enforce the decentralizability constraint of QTRAN. Next, Weighted QMIX use a weighted projection from the true action-value estimator $Q_{\texttt{jt}}$ to the transformed action-value estimator $Q_{\texttt{tran}}$ that allows more emphasis to be placed on better joint actions. We provide a more detailed description of other CTDE algorithms in Appendix B.

## 3  QTRAN++: IMPROVING TRAINING AND ARCHITECTURE OF QTRAN

In this section, we introduce our framework, coined QTRAN++, to decentralize the joint action-value estimator. Similar to the original QTRAN (Son et al., 2019), our approach is based on two types of action-value estimators with different roles. First, we utilize a *true action-value estimator*, which estimates the true action-value using standard temporal difference learning without any compromise for decentralization. On the other hand, we keep a *transformed action-value estimator* for each agent that is decentralizable by construction; they aim at projecting the true action-value estimator into a space of functions that can be decentralized into utility functions.

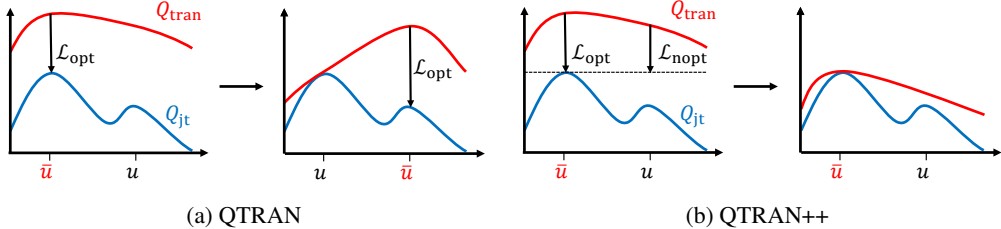

(a) QTRAN        (b) QTRAN++

Figure 2: Learning process of QTRAN and QTRAN++. When $Q_{\texttt{tran}}$ is in overall larger than $Q_{\texttt{jt}}$, QTRAN decreases $Q_{\texttt{tran}}$ only for the optimal action $\bar{\boldsymbol{u}}$. On the other hand, our method efficiently learns the transformed action-value estimator through additional constraint for non-optimal actions.

We further demonstrate our main contribution in the rest of this section. In Section 3.1, we propose a modified training objective of QTRAN, which trains the transformed action-value estimator with enhanced performance. Next, in Section 3.2, we introduce a new architecture for the action-value estimators, modified from QTRAN; we use *semi-monotonic mixing network* for the true action-value estimator and *multi-head monotonic mixing networks* for the transformed action-value estimators.

## 3.1 EFFICIENT TRAINING OF ACTION-VALUE ESTIMATORS

The training objective of QTRAN++ is twofold. First, QTRAN++ trains the true action-value estimator for approximating the true action-value with standard temporal difference learning. Next, the transformed action-value estimators attempt to imitate the behavior of the true action-value estimator. Notably, the utility functions decentralize the transformed action-value estimator by construction, and they can "approximately" decentralize the true action-value estimator when the second objective is sufficiently optimized.

First, we use the loss function $\mathcal{L}_{\texttt{td}}$ for updating the true action-value estimator $Q_{\texttt{jt}}$ with standard temporal difference learning, defined as follows:

$$\mathcal{L}_{\texttt{td}} = \left(Q_{\texttt{jt}}(s, \boldsymbol{\tau}, \boldsymbol{u}) - (r + \gamma Q_{\texttt{jt}}^{\texttt{target}}(s', \boldsymbol{\tau}', \bar{\boldsymbol{u}}'))\right)^2,$$

where $(s, \boldsymbol{\tau}, \boldsymbol{u})$ and $(s', \boldsymbol{\tau}', \bar{\boldsymbol{u}}')$ are objects collected from consecutive time steps in the Markov decision process. Next, $\bar{\boldsymbol{u}}' = [\bar{u}'_1, \dots, \bar{u}'_N]$ is the set of actions maximizing the utility functions, i.e., $\bar{u}'_i = \arg\max_{u_i} q_i(\tau'_i, u'_i)$ for $i \in \mathcal{N}$. Finally, $Q_{\texttt{jt}}^{\texttt{target}}$ is the target network, which periodically updates parameters from $Q_{\texttt{jt}}$.

Next, we demonstrate the training objectives for promoting similar behavior of the true and the transformed action-value estimators. To this end, for each sample $(s, \boldsymbol{\tau}, \boldsymbol{u})$ from the replay buffer, we train one head of the transformed action-value estimator, denoted by $Q_{\texttt{tran}}$, using two loss functions $\mathcal{L}_{\texttt{opt}}$ and $\mathcal{L}_{\texttt{nopt}}$ expressed as follows:

$$\mathcal{L}_{\texttt{opt}}(s, \boldsymbol{\tau}) = \left(Q_{\texttt{jt}}(s, \boldsymbol{\tau}, \bar{\boldsymbol{u}}) - Q_{\texttt{tran}}(s, \boldsymbol{\tau}, \bar{\boldsymbol{u}})\right)^2,$$

$$\mathcal{L}_{\texttt{nopt}}(s, \boldsymbol{\tau}, \boldsymbol{u}) = \begin{cases} (Q_{\texttt{jt}}(s, \boldsymbol{\tau}, \boldsymbol{u}) - Q_{\texttt{tran}}(s, \boldsymbol{\tau}, \boldsymbol{u}))^2, & \text{if } Q_{\texttt{jt}}(s, \boldsymbol{\tau}, \boldsymbol{u}) \geq Q_{\texttt{jt}}(s, \boldsymbol{\tau}, \bar{\boldsymbol{u}}), \\ (Q_{\texttt{clip}}(s, \boldsymbol{\tau}, \boldsymbol{u}) - Q_{\texttt{tran}}(s, \boldsymbol{\tau}, \boldsymbol{u}))^2, & \text{if } Q_{\texttt{jt}}(s, \boldsymbol{\tau}, \boldsymbol{u}) < Q_{\texttt{jt}}(s, \boldsymbol{\tau}, \bar{\boldsymbol{u}}), \end{cases}$$

$$Q_{\texttt{clip}}(s, \boldsymbol{\tau}, \boldsymbol{u}) = \texttt{clip}(Q_{\texttt{tran}}(s, \boldsymbol{\tau}, \boldsymbol{u}), Q_{\texttt{jt}}(s, \boldsymbol{\tau}, \boldsymbol{u}), Q_{\texttt{jt}}(s, \boldsymbol{\tau}, \bar{\boldsymbol{u}})),$$

where $\bar{\boldsymbol{u}}$ is an "optimal" action maximizing the utility functions $q_i$ for $i \in \mathcal{N}$ and the function $\texttt{clip}(\cdot, \ell_1, \ell_2)$ bounds its input to be within the interval $[\ell_1, \ell_2]$.

Similar to QTRAN, our QTRAN++ succeeds in decentralizing the action-value estimator if and only if the training objective is sufficiently minimized. To this end, the two loss functions $\mathcal{L}_{\texttt{opt}}$ and $\mathcal{L}_{\texttt{nopt}}$ enforce the following condition:

$$\underbrace{Q_{\texttt{jt}}(s, \boldsymbol{\tau}, \bar{\boldsymbol{u}})}_{(a)} = \underbrace{Q_{\texttt{tran}}(s, \boldsymbol{\tau}, \bar{\boldsymbol{u}})}_{(b)} > \underbrace{Q_{\texttt{tran}}(s, \boldsymbol{\tau}, \boldsymbol{u})}_{(c)} > \underbrace{Q_{\texttt{jt}}(s, \boldsymbol{\tau}, \boldsymbol{u})}_{(d)}. \tag{4}$$

Note that the utility functions achieve decentralization of the true action-value estimator by definition when (a) > (d). The first loss $\mathcal{L}_{\texttt{opt}}$ enforces (a) = (b) and the second loss $\mathcal{L}_{\texttt{nopt}}$ enforces (a) > (d), (a) > (c), and (c) > (d). See Appendix A for a formal result on this theoretical guarantee.

Table 1: Payoff matrix of the one-step non-decentralizable game and reconstructed results of the game. States $s_1$ and $s_2$ are randomly selected with same probability and the agents cannot observe the state. Boldface means optimal and greedy actions from the state-action value function.

| | A | B | | | A | B | | | A | B | | | A | B |
|---|---|---|---|---|---|---|---|---|---|---|---|---|---|---|
| A | **4** | 2 | | A | 0 | 1 | | A | 3.0 | **3.0** | | A | 0.5 | **1.5** |
| B | 2 | 0 | | B | 1 | **2** | | B | 1.0 | 1.0 | | B | 0.5 | 1.5 |

(a) Payoff of matrix game for state $s_1$ and $s_2$  (b) QMIX: $Q_{\mathtt{jt}}(s_1), Q_{\mathtt{jt}}(s_2)$

| | A | B | | | A | B | | | A | B | | | A | B |
|---|---|---|---|---|---|---|---|---|---|---|---|---|---|---|
| A | **3.0** | 2.0 | | A | **1.0** | 1.0 | | A | **3.0** | 2.5 | | A | **1.0** | 0.5 |
| B | 2.5 | 1.5 | | B | 0.5 | 0.5 | | B | 2.0 | 1.5 | | B | 1.0 | 0.5 |

(c) QTRAN++ head 1: $Q_{\mathtt{tran}}^{(1)}(s_1), Q_{\mathtt{tran}}^{(1)}(s_2)$  (d) QTRAN++ head 2: $Q_{\mathtt{tran}}^{(2)}(s_1), Q_{\mathtt{tran}}^{(2)}(s_2)$

**Comparing with the training objective of $Q_{\mathtt{tran}}$.** QTRAN is similar to our algorithm in a way that it trains the true and the transformed action-value estimators to satisfy Equation 4. However, QTRAN only enforce (a) = (b) and (c) > (d), while our algorithm enforce two additional constraints of (a) > (d) and (a) > (c). As a result, our algorithm trains the estimators with denser training signal and results in a more efficient training. See Figure 2 for an illustration of such an intuition.

**Non-fixed true action-value estimator.** Notably, we train both the true action-value estimator $Q_{\mathtt{jt}}$ and the transformed action-value estimator $Q_{\mathtt{tran}}$ to imitate each other, using loss functions $\mathcal{L}_{\mathtt{opt}}$ and $\mathcal{L}_{\mathtt{nopt}}$. This is in contrast with QTRAN, which only trains $Q_{\mathtt{tran}}$ to imitate $Q_{\mathtt{jt}}$. This allows the true action-value estimator $Q_{\mathtt{jt}}$ to additionally minimize the decentralization error, i.e., the gap between maximizing $Q_{\mathtt{jt}}$ and maximizing agent-wise utility functions $q_i$. Such a modification allows the agents to learn a better policy for maximizing the true action-values.

### 3.2 MIXING NETWORK ARCHITECTURES FOR ACTION-VALUE ESTIMATORS

Here, we introduce our choice of architectures for the true and the transformed action-value estimators. At a high-level, we construct the estimators using the utility functions $q_1, \ldots, q_N$ parameterized as a deep recurrent Q-network (DRQN, Hausknecht and Stone 2015). Our main contribution in designing the estimators is twofold: (i) using a semi-monotonic mixing network for the true action-value estimator $Q_{\mathtt{jt}}$ and (ii) using a multi-head monotonic mixing network for the transformed action-value estimators $Q_{\mathtt{tran}}^{(i)}$ for $i \in \mathcal{N}$. The role of (i) is to additionally impose a bias on the utility functions to decentralize the true action-value estimator $Q_{\mathtt{jt}}$. Next, we introduce (ii) to prevent the utility functions from overly relying on the underlying state information that is unavailable during execution.

**Semi-monotonic mixing network for $Q_{\mathtt{jt}}$.** We express the true action-value estimator as a mixing network applied to decentralized utility functions $q_1, \ldots, q_N$ as follows:

$$Q_{\mathtt{jt}}(s, \boldsymbol{\tau}, \boldsymbol{u}) = f_{\mathrm{mix}}(q_1, \ldots, q_N; \theta_{\mathtt{jt}}^{(1)}(s)) + f_{\mathrm{mix}}(q_1, \ldots, q_N; \theta_{\mathtt{jt}}^{(2)}(s)),$$

where we omit the input dependency of utility functions for convenience of notation, e.g., we let $q_i$ denote $q_i(\tau_i, u_i)$ for agent $i$. Furthermore, we express the mixing networks as fully connected networks where the parameters $\theta_{\mathtt{jt}}^{(1)}(s), \theta_{\mathtt{jt}}^{(2)}(s)$ are obtained from state-dependent hypernetworks.

Importantly, we constrain the second parameter to be non-negative, i.e., $\theta_{\mathtt{jt}}^{(2)}(s) \geq 0$. Similar to QMIX, this guarantees the second network to be monotonic with respect to the utility functions.

Such a design choice yields a "semi-monotonic" true action-value estimator which can estimate non-monotonic action-value functions while implicitly preferring to behave similarly as the monotonic function. Such a semi-monotonic function is easier for the (monotonic) transformed action-value to estimate, leading to smaller decentralization error. As a result, the individual agents (decentralizing $Q_{\mathtt{tran}}$) will effectively maximize the true action-value function.

**Multi-head mixing network for $Q_{\mathtt{tran}}$.** Now we introduce the multi-head mixing network used for the transformed action-value estimator. To this end, we express the $i$-th head of the transformed action-value estimator $Q_{\mathtt{tran}}^{(i)}$ as follows:

$$Q_{\mathtt{tran}}^{(i)}(s, \boldsymbol{\tau}, \boldsymbol{u}) = q_i + f_{\mathrm{mix}}(q_1, \ldots, q_{i-1}, q_{i+1}, \ldots, q_N; \theta_{\mathtt{tran}}^{(i)}(s)) \quad \forall\, i \in \mathcal{N},$$

where $Q_{\mathtt{tran}}^{(i)}$ denotes the $i$-th head of the transformed action-value estimator and we do not denote the input of the utility functions for the convenience of notation, e.g., we used $q_i$ to denote $q_i(\tau_i, u_i)$. Furthermore, each $\theta_{\mathtt{tran}}^{(i)}(s)$ is non-negative parameters obtained from state-dependent hypernetworks.

Following QMIX, we design the multi-head mixing networks to be monotonic and decentralizable using the utility functions. However, unlike QMIX, i.e., Equation 2, we separately design the $i$-th estimator $Q_{\mathtt{tran}}^{(i)}$ for each agent. This allows the utility function $q_i$ to rely less on the underlying state information $s$, which is not accessible during the execution of the algorithms.

We further elaborate our intuition in Table 1. In the table, we compare QMIX and QTRAN++ for a matrix game where states $s_1$ and $s_2$ are randomly selected with the same probability. This matrix game has optimal actions $(A, A)$ and $(B, B)$ at state $s_1$ and state $s_2$, respectively. If the agents cannot observe the state in the execution phase, the task is non-decentralizable, and the best policy leads to choosing an action $(A, A)$. As shown in Table 1b, QMIX sometimes learns a non-optimal policy, which is caused by the different state-dependent output weights of mixing networks. In contrast, QTRAN++ in Table 1c and 1d learns an individual utility function for the average reward of states. A more detailed discussion is available in Appendix C.

## 4 EXPERIMENTS

### 4.1 EXPERIMENTAL SETUP

We mainly evaluate our method on the Starcraft Multi-Agent Challenge (SMAC) environment (Samvelyan et al., 2019). In this environment, each agent is a unit participating in combat against enemy units controlled by handcrafted policies. In the environment, agents receive individual local observation containing distance, relative location, health, shield, and unit type of other allied and enemy units within their sight range. The SMAC environment additionally assumes a global state to be available during the training of the agents. To be specific, the global state contains information of all agents participating in the scenario. Appendix D contains additional experimental details. We also provide additional experiments on the multi-domain Gaussian squeeze environment (Son et al., 2019) in Appendix F.

We compare our method with six state-of-the-art baselines: QMIX (Rashid et al., 2018), QTRAN (Son et al., 2019), VDN (Sunehag et al., 2018), QPLEX (Wang et al., 2020a), OW-QMIX (Rashid et al., 2020b), and CW-QMIX (Rashid et al., 2020b). We consider SMAC maps with five different scenarios, including scenarios with difficulty levels of easy, hard, and super hard. All the algorithms are trained using two million steps.[1] For evaluation, we run 32 test episodes without an exploration factor for every $10^4$-th time step. The percentage of episodes where the agents defeat all enemy units, i.e., test win rate, is reported as the performance of algorithms. All the results are averaged over five independent runs. We report the median performance with shaded 25-75% confidence intervals.

**Rewarding mechanisms.** We consider two types of scenarios with distinct mechanisms of reward functions. In the first case, the reward is determined only by the enemy unit's health condition, i.e., units receive reward only when the enemy unit is damaged. We coin this case as the *standard scenario* to reflect how this setting is widely used by existing works (Rashid et al., 2018; 2020b). Next, we consider the case where each unit is additionally penalized for being damaged, i.e., units are rewarded for having a "selfish" behavior. This case is denoted as the *negative scenario* since we introduced a negative reward for the penalty.

**Ablation studies.** We also conduct additional ablation studies to validate the effectiveness of each component introduced in our method. Namely, we consider to verify the effect of (a) multi-head mixing network, (b) semi-monotonic mixing network, (c) loss function modified from QTRAN, and (d) using non-fixed true action-value estimator during training. To consider (a), we replace the multi-head mixing network with a single-head mixing network, which we call Mix-QTRAN++. For (b), we compare against FC-QTRAN++, which replaces the mixing network used in the true action-value estimator with a feed-forward network takes the state and the appropriate actions' utilities as input. Next, we analyze (c) through comparing with LB-QTRAN++, which trains the proposed action-value

---

[1] We also report experimental results using more training steps in Appendix E.

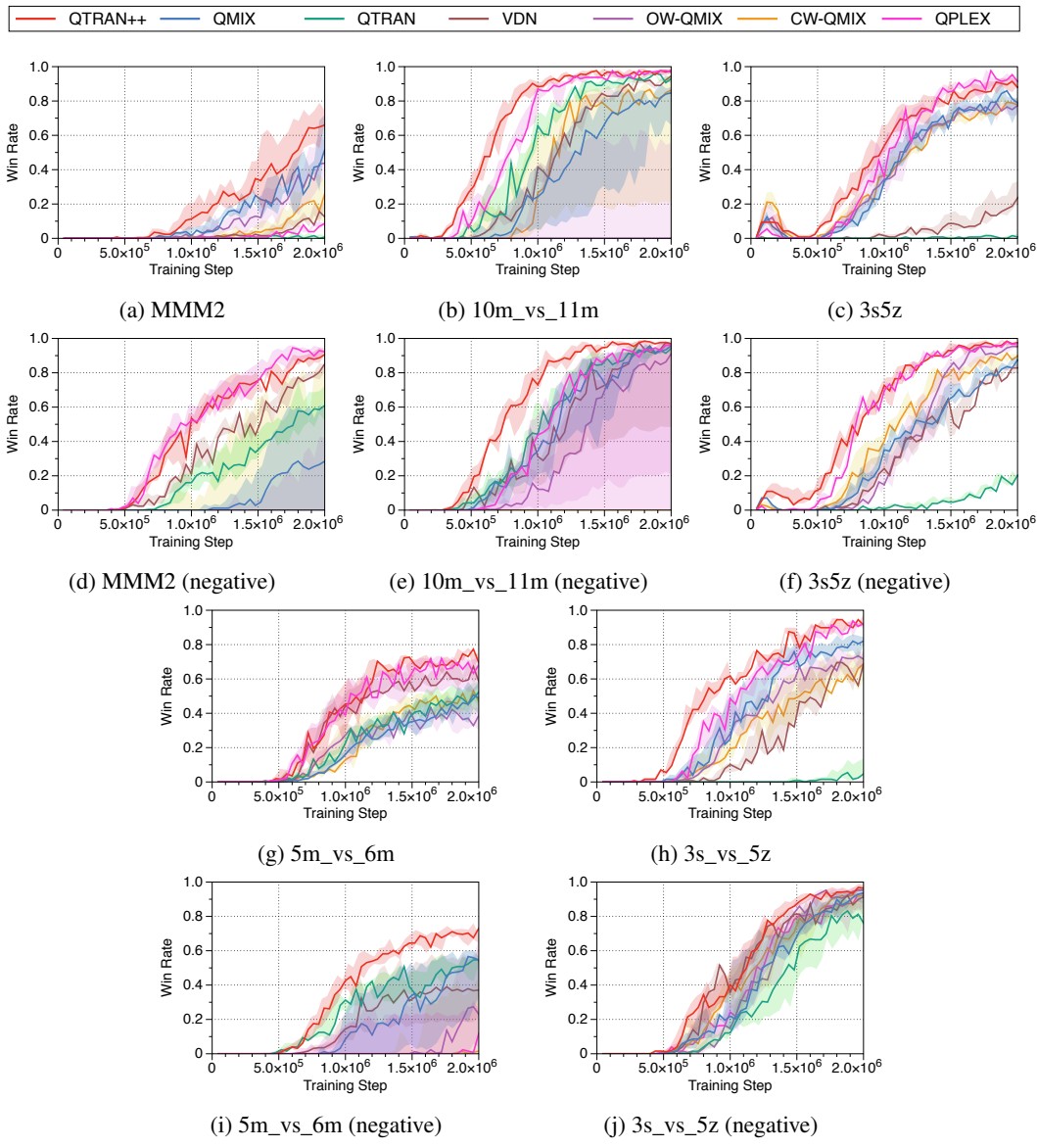

Figure 3: Median test win rate with 25%-75% percentile, comparing QTRAN++ with baselines. All the results are averaged over five independent runs. Each plot corresponds to different scenario names, e.g., 3s5z. Scenarios with additional negative rewarding mechanism are indicated by "(negative)" in the name of scenarios, e.g., 3s5z (negative).

estimators using the loss function of QTRAN. Finally, we validate (d) through comparing with Fix-QTRAN++ which fix the true action-value estimator for optimizing $\mathcal{L}_{\mathtt{opt}}$ and $\mathcal{L}_{\mathtt{nopt}}$.

## 4.2 RESULTS

**Comparisons with baselines.** We report the performance of QTRAN++ and the baseline algorithms in Figure 3. Overall, QTRAN++ achieves the highest win rate compared to existing baselines, regardless of the scenario being standard or negative. Especially, one observes a significant gap between QTRAN++ and the second-best baselines for scenarios such as 10m_vs_11m, 10m_vs_11m (negative), and 5m_vs_6m (negative). This is more significant since there is no consistent baseline achieving the second place. Furthermore, the original version of our algorithm, i.e., QTRAN, performs poorly in several tasks, e.g., Figure 3a and Figure 3c. From such results, one may conclude

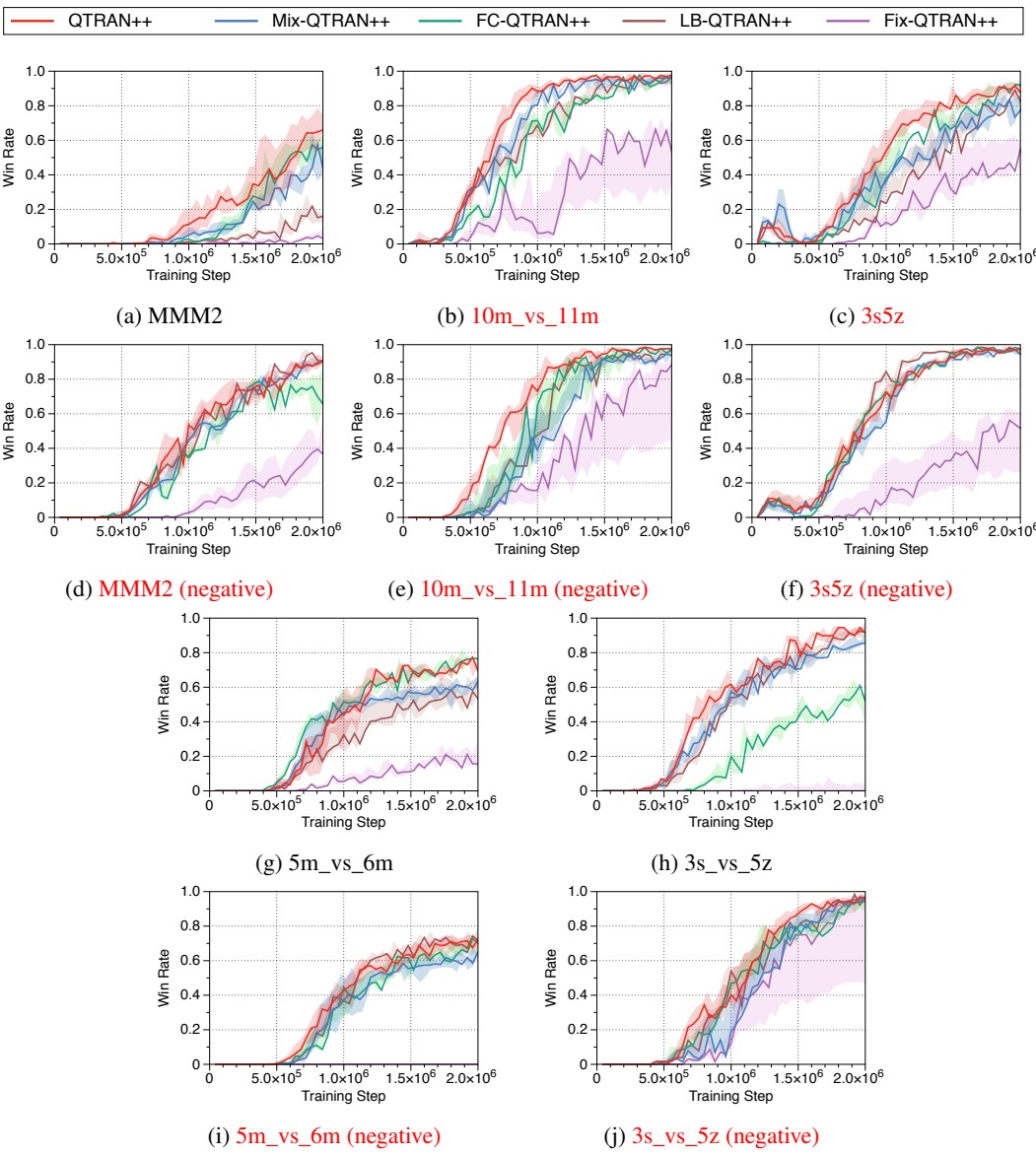

Figure 4: Median test win rate with 25%-75% percentile, comparing QTRAN++ with its modifications. All the results are averaged over five independent runs. Each plot corresponds to different scenario names, e.g., 3s5z. Scenarios with additional negative rewarding mechanism are indicated by "(negative)" in the name of scenarios, e.g., 3s5z (negative).

that QTRAN++ successfully resolves the issue existing in the original QTRAN and achieves state-of-the-art performance compared to baselines for the SMAC environment.

**Observations.** Intriguingly, in Figure 3, one observes an interesting result that negative rewards can improve the test win rate of MARL algorithms. For example, when comparing Figure 3d and Figure 3i, one may observe how the presence of negative rewards speed up the learning process of algorithms. We conjecture our modified environments provide a denser reward signal, enabling the agent to learn policies efficiently.

We also illustrate how QTRAN++ successfully solves the most difficult negative scenarios in Figure 5. In such scenarios, as demonstrated in Figure 5b and Figure 5d, existing algorithm such as QMIX may end up learning "locally optimal" policies where the agents run away from the enemy units to avoid receiving negative rewards. However, in Figure 5a and Figure 5c, one observes that QTRAN++ successfully escapes the local optima and chooses to fight the enemies to achieve a higher win rate.

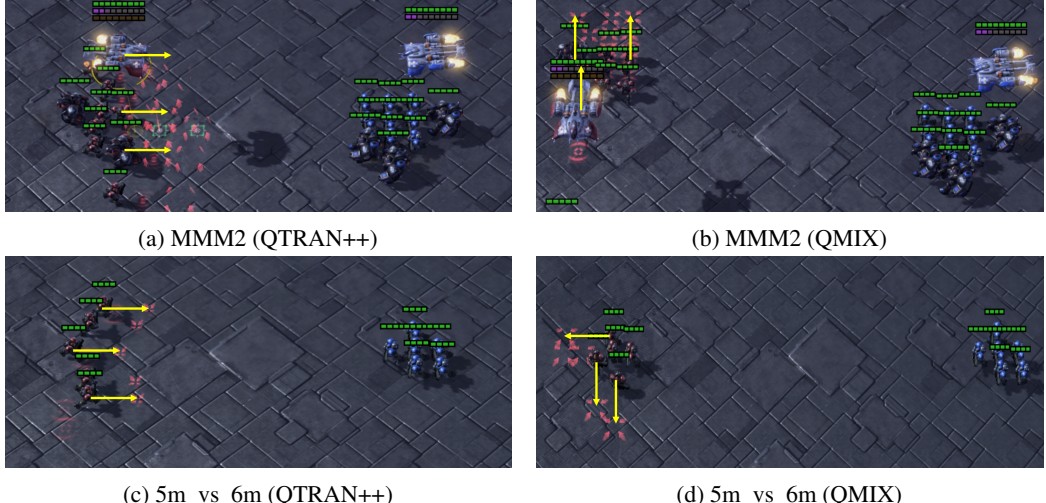

(a) MMM2 (QTRAN++)          (b) MMM2 (QMIX)

(c) 5m_vs_6m (QTRAN++)          (d) 5m_vs_6m (QMIX)

Figure 5: Illustration of the policies learned using QTRAN++ and QMIX in negative scenarios. Agents trained by QMIX run away from enemies without fighting (b, d). In contrast, QTRAN++ trains the agents to fight the enemies, obtaining a higher win rate in overall (a, c).

**Ablation study.** We show the results for our ablation study in Figure 4. We consider three scenarios to show how each element of QTRAN++ is effective for improving its performance. Overall, one can observe how QTRAN++ demonstrates a solid improvement over its ablated versions. In particular, QTRAN++ outperforms them by a large margin in at least one of the considered scenarios. For example, QTRAN++ significantly outperforms FC-QTRAN++ in Figure 4a and 4h. It also significantly outperforms Mix-QTRAN++, LB-QTRAN++, Fix-QTRAN++ in Figure 4a and 4g. Such a result validates how the algorithmic components in our algorithm are complementary to each other and crucial for consistently achieving high empirical performance.

## 5    CONCLUSION

In this paper, we present QTRAN++, a simple yet effective improvement over QTRAN (Son et al., 2019) for cooperative multi-agent reinforcement learning under the paradigm of centralized training with decentralized execution. Our gains mainly come from (i) stabilizing the training objective of QTRAN, (ii) removing the strict role separation between the action-value estimators, and (iii) introducing a multi-head mixing network for value transformation. Using the Starcraft Multi-Agent Challenge (SMAC) environment, we empirically show how our method improves QTRAN and achieves state-of-the-art results.

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

# A PROOFS

**Theorem 1.** *There exists a set of utility functions $\{q_i\}_{i \in \mathcal{N}}$ decentralizing the true action-value estimator $Q_{\mathrm{jt}}(s, \boldsymbol{\tau}, \boldsymbol{u})$ if and only if there exists a function $Q'_{\mathrm{jt}}(s, \boldsymbol{u})$ satisfying the following conditions:*

$$Q_{\mathrm{tran}}(s, \boldsymbol{\tau}, \bar{\boldsymbol{u}}) = Q_{\mathrm{jt}}(s, \boldsymbol{\tau}, \bar{\boldsymbol{u}}), \tag{5}$$

$$Q_{\mathrm{tran}}(s, \boldsymbol{\tau}, \boldsymbol{u}) \geq Q_{\mathrm{jt}}(s, \boldsymbol{\tau}, \boldsymbol{u}), \tag{6}$$

$$\frac{\partial Q_{\mathrm{tran}}(s, \boldsymbol{\tau}, \boldsymbol{u})}{\partial q_i(\tau_i, u_i)} \geq 0, \qquad \forall i \in \mathcal{N}, \tag{7}$$

$$\bar{\boldsymbol{u}} = [\arg \max_{u_i} q_i(\tau_i, u_i)]_{i \in \mathcal{N}}. \tag{8}$$

*Proof.* $\Longleftarrow$ We prove first the sufficiency of the theorem by showing that if the conditions hold, then $q_i(\tau_i, u_i)$ satisfies optimal decentralization $\arg \max_{\boldsymbol{u}} Q_{\mathrm{jt}}(s, \boldsymbol{u}) = \bar{\boldsymbol{u}}$.

$$
\begin{aligned}
Q_{\mathrm{jt}}(s, \boldsymbol{\tau}, \bar{\boldsymbol{u}}) &= Q_{\mathrm{tran}}(s, \boldsymbol{\tau}, \bar{\boldsymbol{u}}) &&\text{(From (5))} \\
&\geq Q_{\mathrm{tran}}(s, \boldsymbol{\tau}, \boldsymbol{u}) &&\text{(From (7) and (8))} \\
&\geq Q_{\mathrm{jt}}(s, \boldsymbol{\tau}, \boldsymbol{u}). &&\text{(From (6))}
\end{aligned}
$$

It means that the set of local optimal actions $\bar{\boldsymbol{u}}$ maximizes $Q_{\mathrm{jt}}$, showing that $q_i$ satisfies decentralizability.

$\Longrightarrow$ We turn now to the necessity. First, we define $Q_{\mathrm{tran}}(s, \boldsymbol{\tau}, \boldsymbol{u}) = \sum_{i=1}^{N} \alpha_i(q_i(\tau_i, u_i) - q_i(\tau_i, \bar{u}_i)) + Q_{\mathrm{jt}}(s, \boldsymbol{\tau}, \bar{\boldsymbol{u}})$, which satisfies condition Equation 5 and Equation 7, , where constant $\alpha_i \geq 0$. Since the set of utility functions $\{q_i\}_{i \in \mathcal{N}}$ decentralizes the true action-value estimator $Q_{\mathrm{jt}}$, utility functions and true action-value estimator satisfy the condition $A_{\mathrm{jt}}(s, \boldsymbol{\tau}, \boldsymbol{u}) = Q_{\mathrm{jt}}(s, \boldsymbol{\tau}, \boldsymbol{u}) - Q_{\mathrm{jt}}(s, \boldsymbol{\tau}, \bar{\boldsymbol{u}}) < 0$ if $\boldsymbol{u} \neq \bar{\boldsymbol{u}}$. So proof for Equation 6 follows from the fact that there exists $[\alpha_i]$ small enough such that

$$Q_{\mathrm{tran}}(s, \boldsymbol{\tau}, \boldsymbol{u}) - Q_{\mathrm{jt}}(s, \boldsymbol{\tau}, \boldsymbol{u}) = \sum_{i=1}^{N} \alpha_i(q_i(\tau_i, u_i) - q_i(\tau_i, \bar{u}_i)) - A_{\mathrm{jt}}(s, \boldsymbol{\tau}, \boldsymbol{u})) \geq 0.$$

$\square$

# B  RELATED WORK

Centralized training with decentralized execution (CTDE) has emerged as a popular paradigm under the multi-agent reinforcement learning framework. It assumes the complete state information to be fully accessible during training, while individual policies allow decentralization during execution. To train agents under the CTDE paradigm, both policy-based (Foerster et al., 2018; Lowe et al., 2017; Du et al., 2019; Iqbal and Sha, 2019; Wang et al., 2020b) and value-based methods (Sunehag et al., 2018; Rashid et al., 2018; Son et al., 2019; Yang et al., 2020) have been proposed. At a high-level, the policy-based methods rely on the actor-critic framework with independent actors to achieve decentralized execution. On the other hand, the value-based methods attempt to learn a joint action-value estimator, which can be cleverly decomposed into individual agent-wise utility functions.

For examples of the policy-based methods, COMA (Foerster et al., 2018) trains individual policies with a joint critic and solves the credit assignment problem by estimating a counterfactual baseline. MADDPG (Lowe et al., 2017) extends the DDPG (Lillicrap et al., 2015) algorithm to learn individual policies in a centralized manner on both cooperative and competitive games. MAAC (Iqbal and Sha, 2019) includes an attention mechanism in critics to improve scalability. LIIR (Du et al., 2019) introduces a meta-gradient algorithm to learn individual intrinsic rewards to solve the credit assignment problem. Recently, ROMA (Wang et al., 2020b) proposes a role-oriented framework to learn roles via deep RL with regularizers and role-conditioned policies.

Among the value-based methods, value-decomposition networks (VDN, Sunehag et al. 2018) learns a centralized, yet factored joint action-value estimator by representing the joint action-value estimator as a sum of individual agent-wise utility functions. QMIX (Rashid et al., 2018) extends VDN by employing a *mixing network* to express a non-linear monotonic relationship among individual agent-wise utility functions in the joint action-value estimator. Qatten (Yang et al., 2020) introduces a multi-head attention mechanism for approximating the decomposition of the joint action-value estimator, which is based on theoretical findings.

QTRAN (Son et al., 2019) has been proposed recently to eliminate the monotonic assumption on the joint action-value estimator in QMIX (Rashid et al., 2018). Instead of directly decomposing the joint action-value estimator into utility functions, QTRAN proposes a training objective that enforces the decentralization of the joint action-value estimator into the summation of individual utility functions. In the QTRAN paper, a new algorithm called QTRAN-alt is also proposed to more accurately distinguish the transformed action-value for optimal actions from non-optimal actions. However, this QTRAN-alt algorithm has high computational complexity because it requires counterfactual action-value estimation when other actions are selected. In addition, Mahajan et al. (2019) experimentally shows that QTRAN-alt does not work as well as QTRAN-base in StarCraft Multi-Agent Challenge (Samvelyan et al., 2019) environments. In this paper, we only deal with QTRAN-base modifications, and our proposed method still has the advantage of QTRAN-alt without counterfactual action-value estimation.

Recently, several other methods have been proposed to solve the limitations of QMIX. QPLEX (Wang et al., 2020a) takes a duplex dueling network architecture to factorize the joint value function. Unlike QTRAN, QPLEX learns using only one joint action-value network and a single loss function. Their experiments on the StarCraft Multi-Agent Challenge (SMAC) environment (Samvelyan et al., 2019) also mainly utilized offline data. Finally, Rashid et al. (2020b) proposed CW-QMIX and OW-QMIX which use a weighted projection that allows more emphasis to be placed on better joint actions. However, their methods are susceptible to weight parameter changes and the performance improvements have been demonstrated in limited SMAC environments only. Unlike these methods, we propose novel and simple-to-implement modifications that substantially improve the stability and performance of the original QTRAN.

## C    ONE-STEP NON-DECENTRALIZABLE MATRIX GAME

Table 2: Payoff matrix of the one-step non-decentralizable game and reconstructed results of the game. States $s_1$ and $s_2$ are randomly selected with same probability and the agents cannot observe the state. Boldface means optimal and greedy actions from the state-action value function.

|   | A | B |   |   | A | B |
|---|---|---|---|---|---|---|
| A | **4** | 2 | | A | 0 | 1 |
| B | 2 | 0 | | B | 1 | **2** |

(a) Payoff of matrix game for state $s_1$ and $s_2$

|   | A | B |   |   | A | B |
|---|---|---|---|---|---|---|
| A | **2.0** | 1.5 | | A | **2.0** | 1.5 |
| B | 1.5 | 1.0 | | B | 1.5 | 1.0 |

(b) VDN: $Q_{\mathtt{jt}}$

|   | A | B |   |   | A | B |
|---|---|---|---|---|---|---|
| A | **4.0** | 2.0 | | A | **1.0** | 1.0 |
| B | 2.0 | 0.0 | | B | 1.0 | 1.0 |

(c) QMIX case 1: $Q_{\mathtt{jt}}(s_1), Q_{\mathtt{jt}}(s_2)$

|   | A | B |   |   | A | B |
|---|---|---|---|---|---|---|
| A | 3.0 | **3.0** | | A | 0.5 | **1.5** |
| B | 1.0 | 1.0 | | B | 0.5 | 1.5 |

(d) QMIX case 2: $Q_{\mathtt{jt}}(s_1), Q_{\mathtt{jt}}(s_2)$

|   | A | B |   |   | A | B |
|---|---|---|---|---|---|---|
| A | **3.0** | 2.0 | | A | **1.0** | 1.0 |
| B | 2.5 | 1.5 | | B | 0.5 | 0.5 |

(e) QTRAN++ head 1: $Q_{\mathtt{tran}}^{(1)}(s_1), Q_{\mathtt{tran}}^{(1)}(s_2)$

|   | A | B |   |   | A | B |
|---|---|---|---|---|---|---|
| A | **3.0** | 2.5 | | A | **1.0** | 0.5 |
| B | 2.0 | 1.5 | | B | 1.0 | 0.5 |

(f) QTRAN++ head 2: $Q_{\mathtt{tran}}^{(2)}(s_1), Q_{\mathtt{tran}}^{(2)}(s_2)$

This section presents how multi-head mixing networks perform to existing methods such as VDN and QMIX. The matrix game and learning results are shown in Table 2. In this matrix game, states $s_1$ and $s_2$ are randomly selected with the same probability. This matrix game has the optimal action $(A, A)$ at $s_1$, and $(B, B)$ at $s_2$. If the agents cannot observe the state, the task is non-decentralizable and the best policy that agents can do is to choose an action $(A, A)$ that can get the highest rewards on average. Now we show the results of VDN, QMIX, and QTRAN++ through a full exploration. First, Table 2b shows that VDN enables each agent to jointly take the best action only by using its own locally optimal action despite not learning the correct action-values. Table 2c also demonstrates that QMIX learns the best policy. However, as shown in Table 2d, QMIX sometimes learns a non-optimal policy, which is caused by the state-dependent output weights of mixing networks. VDN always learns individual utility functions for the average reward of states, but QMIX tends to make the zero mixing network weights for one of the two states to reduce TD-error, and which state to ignore depends on the initial values.

Table 2e and 2f show how multi-head mixing network solves the biased credit assignment problem with the transformed action-value estimator. Each head $i$ of our method assigns the unbiased credit for agent $i$, allowing it to learn the best policy. Multi-head mixing networks do not perfectly maintain the representation power of the original mixing network. But as the number of agents increases, the representation power between the two methods becomes the same.

# D    EXPERIMENTAL DETAILS

The hyperparameters of training and testing configurations for VDN, QMIX, QTRAN are the same as in the recent GitHub code of SMAC (Samvelyan et al., 2019) with StarCraft version SC2.4.10. The architecture of all agents' policy networks is a DRQN consisting of two 64-dimensional fully connected layers and 64-dimensional GRU. The mixing networks consist of a single hidden layer with 32 hidden widths and ELU activation functions. Hypernetworks consist of two layers with 64 hidden widths and ReLU activation functions. For Weighted QMIX (Rashid et al., 2020b), we additionally use a feed-forward network with 3 hidden layers of 128 dim and ReLU non-linearities and set $\alpha = 0.75$. Finally, the hyperparameters of QPLEX are the same as in their GitHub code.

All neural networks are trained using the RMSProp optimizer with 0.0005 learning rates. We use $\epsilon$-greedy action selection with decreasing $\epsilon$ from 1 to 0.05 over 500000 time steps for exploration, following Rashid et al. (2020b). For the discount factor, we set $\gamma = 0.99$. The replay buffer stores 5000 episodes at most, and the maximum size of mini-batch is 32. Using a Nvidia Titan Xp graphic card, the training time varies from 8 hours to 24 hours for different scenarios.

In our implementation, all heads share the monotonic mixing network, which takes individual utility functions as input. For head $i$, the mixing network receives the action-independent value $v_i(s)$ as an input instead of the utility function $q_i(\tau_i, u_i)$. Gradients are propagated into all utility functions for training each $i$-th head. This structure makes it possible to distinguish multi-head values with only one mixing network efficiently, and all transformed action-value estimators share the same optimal action. Furthermore, we use state-independent weights for tasks with heterogeneous agents. For the loss function $\mathcal{L}_{\mathrm{nopt}}$, we fixed the value only for the threshold $\widehat{Q}_{\mathrm{jt}}(s, \boldsymbol{\tau}, \bar{\boldsymbol{u}})$ of the clipping function that receives optimal actions as input. To combine our loss functions, we obtain the following objective, which is minimized in an end-to-end manner to train the true action-value estimator and the transformed action-value estimator:

$$\mathcal{L} = \mathcal{L}_{\mathrm{td}} + \lambda_{\mathrm{opt}}\mathcal{L}_{\mathrm{opt}} + \lambda_{\mathrm{nopt}}\mathcal{L}_{\mathrm{nopt}}$$

where $\lambda_{\mathrm{opt}}, \lambda_{\mathrm{nopt}} > 0$ are hyperparameters controlling the importance of each loss function. We set $\lambda_{\mathrm{opt}} = 2$ and $\lambda_{\mathrm{nopt}} = 1$.

## E    EXPERIMENTS USING MORE TRAINING STEPS

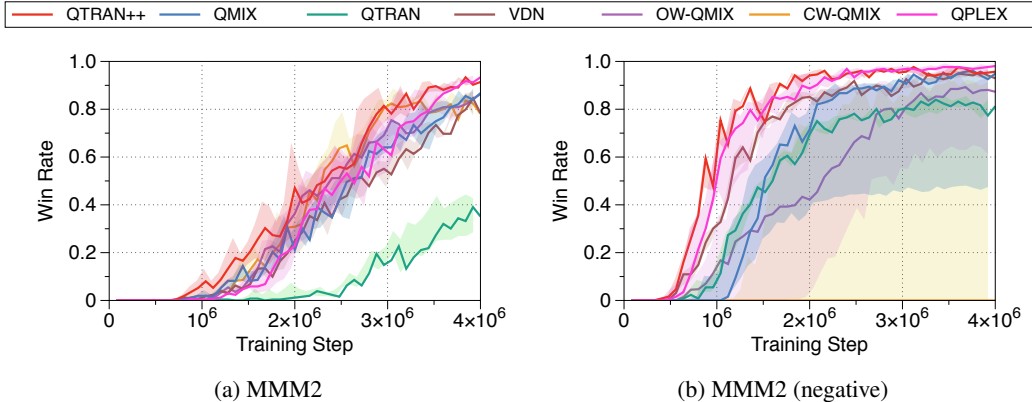

(a) MMM2                                      (b) MMM2 (negative)

Figure 6: Median test win rate with 25%-75% percentile, comparing QTRAN++ to baselines using more training steps. All the results are averaged over five independent runs.

In this section, we provide additional experimental results on the SMAC environment, where we train algorithms using more training steps than that of Figure 4. Our goal is to verify whether if our QTRAN++ maintains state-of-the-art performance even when the algorithms have converged. To this end, we choose the scenarios where QTRAN++ and the baselines have not converged in Figure 4, i.e., MMM2 and MMM2 (negative). Next, we evaluate QTRAN++ and the baselines using four million training steps. The corresponding results are reported in Figure 6. Here, one can observe how QTRAN++ still maintains state-of-the-art performance in both scenarios, regardless of the number of training steps used.

## F    MULTI-DOMAIN GAUSSIAN SQUEEZE EXPERIMENT

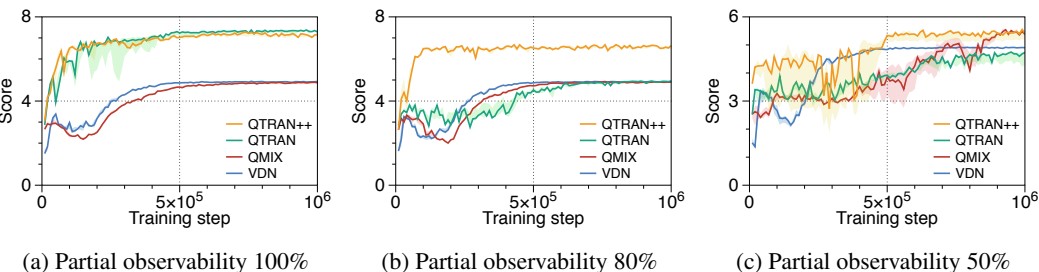

(a) Partial observability 100%    (b) Partial observability 80%    (c) Partial observability 50%

Figure 7: Experimental result on the multi-domain Gaussian squeeze environment. Partial observability indicates the amount of observation provided for each agent.

In this section, we provide the experimental results for the multi-domain Gaussian squeeze (MGS) environment proposed by Son et al. (2019). The MGS environment is specially designed for demonstrating the ability of QTRAN to learn non-monotonic action-value estimators; it modifies the Gaussian squeeze environment (HolmesParker et al., 2014) to have multiple agent-wise policies that achieve locally optimal performance. We evaluate QTRAN++ and other baselines under MGS environment with ten agents. Furthermore, we vary the amount of observation provided for the agents to evaluate the performance of algorithms to consider more challenging scenarios. Note that the MGS environment originally assumes the agents have full observation on the environment.

In Figure 7, one can observe that QTRAN++ consistently achieves the best result compared to the baselines. Both QTRAN and QTRAN++ demonstrates the highest performance in the fully observable environment. However, the performance of QTRAN dramatically decreases as the observation of agent gets more limited. Hence, one may conclude that QTRAN++ achieves state-of-the-art performance across different benchmarks and still retains the main strengths of QTRAN.

# G   ADDITIONAL EXPERIMENTS ON THE SEMI-MONOTONIC MIXING NETWORK

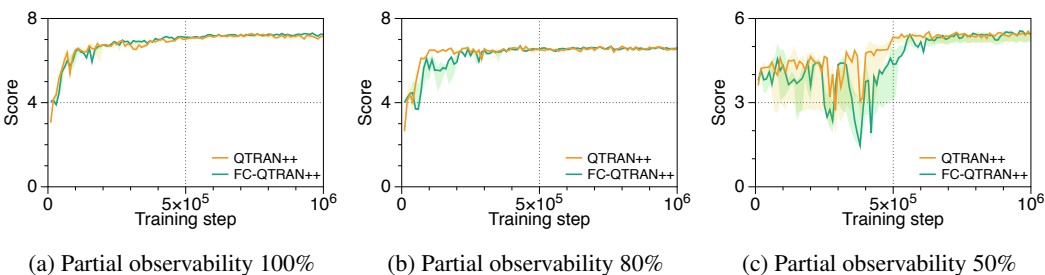

(a) Partial observability 100%         (b) Partial observability 80%         (c) Partial observability 50%

Figure 8: Experimental result on the multi-domain Gaussian squeeze environment. Partial observability indicates the amount of observation provided for each agent.

In this section, we provide additional ablation studies to validate how our semi-monotonic mixing network provides a consistent improvement even on environments where monotonic action-value estimators fail. To this end, we consider comparing QTRAN++ and FC-QTRAN++ in the mult-domain Gaussian squeeze (MGS) environment (Son et al., 2019): a complex multi-agent resource allocation problem that requires learning a non-monotonic action-value estimator. Especially, Son et al. (2019) reported that algorithms that relies on a monotonic action-value estimator, i.e., VDN and QMIX, to perform much worse than algorithms that use a non-monotonic action-value estimator, i.e., QTRAN. In Figure 8, one observes that QTRAN++ performs at least as good as FC-QTRAN++, verifying that our semi-monotonic mixing network is useful regardless of a non-monotonic action-value estimator being required or not.

