# OpenReview forum: "QTRAN++: Improved Value Transformation for Cooperative Multi-Agent Reinforcement Learning"
_ICLR.cc/2021/Conference — Reject_

### Official Review · AnonReviewer2 · 2020-10-27
**QTRAN++ is a good improvement for QTRAN, but the expectation of the advanced version of QTRAN is higher than in 2019.**

**Rating:** 4
**Confidence:** 4

**Review:**

This paper provides good improvements that make QTRAN more practical and can be applied to problems other than matrix games. Since QTRAN is a significant improvement for value-based multi-agent reinforcement learning after QMIX, the practical implementation of QTRAN is expected for a long time in the community. However, after the publication of QTRAN, especially in 2020, some other works have explored the question of how to extend QMIX to the full IGM function class. Due to these works (QPLEX is the major concern, given that weighted-QMIX has been compared in the experiments), the expectation of the advanced version of QTRAN is higher than before. I have several concerns regarding several core contributions of QTRAN++.

QTRAN++ relies heavily on the true joint action-value function. (1.1) However, learning joint action-value functions is not an adorable choice in multi-agent problems. (1.2) To ease the training and representation of joint action-value functions, the authors condition $Q_{jt}$ on individual q values and use a semi-monotonic structure. However, it is difficult to tell the contribution of the monotonic part. It has been shown that monotonic functions can not represent some Q-values. Why should this part be included? I expect that I can find the answer from ablation studies, but on two out of three scenarios, FC-QTRAN++ is very similar to QTRAN++. The authors can provide a more serious discussion of this part to make their paper stronger.

About the training of $Q_{tran}^{i}$. I have two questions about the training of this value function. (2.1) When training $Q_{tran}^{i}$, whether local utility function of agent $j$ ($q_j$ using the notation from the paper) is updated? (2.2) The training scheme is a midpoint between VDN and QMIX, which is similar to an attention mechanism that has been explored in multi-agent value decomposition settings. The formulation is quite different from previous papers (DOP [Wang et al., 2020] and REFIL [Iqbal et al. 2020]), but based on the results from these previous work, I think the multi-head structure may not improve the performance. Although the authors use a matrix game to illustrate their idea, which I appreciate, I can hardly tell whether this example is specially designed. I was expecting a convincing ablation study on SMAC, but I do not find them sufficient: (1) The authors did not record how many random seeds did they test, and SMAC tasks are typically sensitive to random seeds. (2) The gap between QTRAN and QTRAN++ is not significant. If the authors can provide results with more random seeds on more maps, I will consider revising my rating.

My last concern is about QPLEX, as cited by the authors. Similar to QTRAN, QPLEX provides full expressivity for the IGM function class. Nevertheless, the implementation of QPLEX seems to be much more lightweight than QTRAN++. Since QPLEX has provided codes that can be freely tested on the SMAC benchmark, I was wondering why the authors cited this paper but did not compare to it. At least, a detailed discussion of the differences can make the contribution of QTRAN++ clearer.

** A minor concern about experiments.
It seems that the authors are using an older version of QMIX. In the latest version, QMIX can achieve a win rate pf 80% on MMM2. This fact is unknown for many, because the journal version of QMIX reports the same win rate as in this paper.

[Wang et al., 2020] Wang, Y., Han, B., Wang, T., Dong, H. and Zhang, C., 2020. Off-Policy Multi-Agent Decomposed Policy Gradients. arXiv preprint arXiv:2007.12322.

[Iqbal et al. 2020] Iqbal, S., de Witt, C.A.S., Peng, B., Böhmer, W., Whiteson, S. and Sha, F., 2020. AI-QMIX: Attention and Imagination for Dynamic Multi-Agent Reinforcement Learning. arXiv preprint arXiv:2006.04222.



##### ======================== UPDATE =========================

Thanks for the authors' clarifications.

After a careful re-evaluation of the paper, I have many concerns about the performance of baselines on the StarCraft II benchmark tasks. The reported performance is not consistent with those reported in the SMAC benchmark paper (see Figure 4,5,6 in [1]) and QPLEX paper (Figure 5,8,19 in [2]). Moreover, I also evaluate the available GitHub codes of baselines on my own, which is consistent with [1,2].

Using results in [1,2], QTRAN++ significantly underperforms the baselines on the StarCraft II benchmark tasks. Moreover, the paper claims that it uses the standard StarCraft II benchmark, the latest version of SC2, and the default baseline codes.

Due to these concerns, I tend to lower my rating.

[1] Samvelyan M, Rashid T, de Witt C S, et al. The starcraft multi-agent challenge. arXiv preprint arXiv:1902.04043, 2019.

[2] Jianhao Wang, Zhizhou Ren, Terry Liu, Yu Yang, and Chongjie Zhang. Qplex: Duplex dueling multi-agent Q-learning. ICLR submission. https://openreview.net/forum?id=Rcmk0xxIQV

---

> ### Author Response · Authors · 2020-11-20
> **Response to R2 (2/2)**
>
>
>
>
> **6. The ablation studies are not sufficient (for validating the contribution of the multi-head structure). The authors did not record how many random seeds they test. Authors can provide results with more random seeds on more maps.**
>
> Thank you for the suggestion. To address your concern, we extended our ablation studies from 3 maps to 10 maps. We report the experimental results in Figure 4 of the revised paper. In the figure, one can observe how the multi-head component provides a concrete improvement to QTRAN++. To be specific, QTRAN++ performs significantly better than Mix-QTRAN++ (QTRAN++ without the multi-head component) in the 5m_vs_6m and 5m_vs_6m (negative) scenarios and performs at least as good as Mix-QTRAN++ for the other eight scenarios.
>
> Furthermore, we recorded that we use five random seeds for all the experiments in Section 4.1 of our paper (before revision). If you find this number of random seeds insufficient, we will be happy to run more experiments and increase the number of random seeds.
>
> ---
>
> **7. The gap between QTRAN and QTRAN++ is not significant.**
>
> We do believe that the gap between QTRAN and QTRAN++ is significant. As one can see in Figure 3 of our (original or revised) paper, QTRAN++ outperforms QTRAN for all the scenarios. Especially, QTRAN performs the worst among the baselines in five out of ten scenarios, i.e., MMM2, 3s5z, 3s5z (negative), 3s_vs_5z, 3s_vs_5z (negative). In contrast, QTRAN++ achieves the best performance for all ten scenarios.
>
> ---
>
> **8. I was wondering why the authors did not compare with QPLEX. A detailed discussion of the differences between QPLEX and QTRAN++ can make the contribution clearer.**
>
> Thank you for the suggestion. To address your concern, we additionally considered QPLEX (a concurrent submission at ICLR 2021, https://openreview.net/forum?id=Rcmk0xxIQV) as a baseline in Figure 3. One can observe how our QTRAN++ outperforms QPLEX for four scenarios, i.e., MMM2, 10m_vs_11m (negative), 5m_vs_6m (negative), 3z_vs_5z, and performs at least as good as QPLEX for the other six scenarios.
>
> To further incorporate your suggestion, we discuss the algorithmic difference between QPLEX and QTRAN++. As you mentioned, QPLEX is similar to QTRAN++ since it removes the restriction on the true action-value estimator. However, they use different regularization for achieving this goal. To be specific, QTRAN++ imposes an additional loss function between the true and the transformed action-value estimators. In contrast, QPLEX introduces a structural constraint: the true action-value estimator is expressed as a summation of utility functions and a non-positive advantage estimator. A more detailed discussion is incorporated in Appendix B of our revised paper.
>
> ---
>
> **9. It seems that the authors are using an older version of QMIX. In the latest version, QMIX can achieve a win rate of 80% on MMM2.**
>
> After careful inspection, we checked that our version of QMIX is identical to your suggested version. We hypothesize that the performance gap comes from different exploration hyperparameters and version of Starcraft II (performance of the algorithms may be sensitive to the version of Starcraft II as stated in [7]). To be specific, we use the same experimental setting across all algorithms for a fair comparison. Especially, we use the setting employed by the previous state-of-the-art work [4] to make the comparison more competitive. To incorporate your comment, we revised Appendix D of our paper by adding this discussion accordingly.
>
> ---
>
> **References**
> [1] Sunehag et al. Value-Decomposition Networks For Cooperative Multi-Agent Learning, AAMAS 2018
> [2] Rashid et al. Monotonic Value Function Factorisation for Deep Multi-Agent Reinforcement Learning, JMLR 2020
> [3] Son et al. Qtran: Learning to factorize with transformation for cooperative multi-agent reinforcement learning, ICML 2019
> [4] Rashid et al. Weighted QMIX: Expanding Monotonic Value Function Factorisation, NeurIPS 2020
> [5] Wang et al. Off-Policy Multi-Agent Decomposed Policy Gradients, preprint 2020
> [6] Iqbal et al. AI-QMIX: Attention and Imagination for Dynamic Multi-Agent Reinforcement Learning, preprint 2020
> [7] Python MARL framework (https://github.com/oxwhirl/pymarl)

---

> > ### Comment · AnonReviewer2 · 2020-11-23
> > **Re: Response**
> >
> > I want to thank the authors for their careful revision and considerate response.
> >
> > I discussed the paper with some other researchers in the MARL community, and I think a common concern is about the semi-monotonic joint Q network. I am still unclear about the monotonic part. The authors clarify this can "bias the true action-value estimator towards being learned easily by a monotonic function". I didn't quite get the point. Is this generally good for all tasks apart from SC2 (explained in detail in the next paragraph)?
> >
> > In the revised paper, I think the authors mean that, for those action-value functions in the monotonic class, such a structure may learn efficiently. Authors show more experimental results to show the superiority of the semi-monotonic structure, which I appreciate. However, this raises a further concern: whether the proposed structure bypasses the difficulty of estimating joint Q values by exploiting the fact that monotonic functions are sufficient for most maps in the SMAC benchmark? (some recent papers support this fact) The monotonic part may hurt performance on other tasks like Predator and Prey.

---

> > > ### Author Response · Authors · 2020-11-23
> > > **Thanks for your response!**
> > >
> > > Thank you very much for sharing the concern.
> > >
> > > We first clarify your statement: our proposal to use a semi-monotonic structure is not to **learn** the action-value functions efficiently in the monotonic class (for the given environment). Instead, we designed our semi-monotonic network for being efficiently **learned by** the (transformed) action-value functions in the monotonic class.
> > >
> > > We also verify that the semi-monotonic network does not over-exploit the fact that monotonic functions are sufficient for most maps in the SMAC benchmark. To this end, we refer to Figure 4 of our revised paper; here, our QTRAN++ outperforms its variant without the semi-monotonic network (FC-QTRAN++) in the maps where the monotonic functions are not sufficient, i.e., MMM2 (negative) and 10m_vs_11m. We consider the monotonic functions to be insufficient for such maps since the (monotonic) QMIX performs considerably worse than the (non-monotonic) QTRAN for solving them in Figure 3.
> > >
> > > To further alleviate your concern on the semi-monotonic network, we considered an additional task, i.e., multi-domain Gaussian squeeze (MGS) benchmark [8], to compare QTRAN++ and its variant without the semi-monotonic network (FC-QTRAN++). The MGS benchmark is meaningful since it is specifically designed to show the case where the monotonicity functions are insufficient, e.g., QMIX performs worse than QTRAN. We report the corresponding result in Figure 8 of the revised paper. Here, one observes how QTRAN++ performs at least as well as FC-QTRAN++. This again verifies how our semi-monotonic network does not hurt the performance of QTRAN++ for general tasks.
> > >
> > > **References**
> > > [8] Son et al. QTRAN: Learning to factorize with transformation for cooperative multi-agent reinforcement learning, ICML 2019

---

> ### Author Response · Authors · 2020-11-20
> **Response to R2 (1/2)**
>
> We express our deep appreciation for your time and insightful comments. We are grateful for all the positive comments: providing a practical improvement for the theoretically important algorithm (by you and R4), strong empirical performance (by R1, R3, and R4), novel and general ideas (by R1), and clear writing (by R1 and R3). In the revised manuscript, we have substantially updated or newly added (Section 2, Section 3, Figure 3, Figure 4, Appendix B, D, E, F) according to the initial reviews and colored them red. In the following, we address your comments one by one.
>
> ---
>
> **1. The true joint action-value estimator is not an adorable choice in multi-agent problems.**
>
> We agree that using the true joint action-value estimator is not adorable for some multi-agent frameworks, e.g., VDN [1] or QMIX [2], since the agent-wise policies cannot be extracted from the estimator. However, as we explained in Section 3, our QTRAN++ resolves this issue by extracting the agent-wise policies from the transformed action-value estimator: a projection of the true joint action-value estimator into a space of decentralizable functions. Note that QTRAN [3] and Weighted QMIX [4] also used the same approach to use the true joint action-value estimator. In our revised paper, we further clarified this point at the beginning of Section 3.
>
> ---
>
> **2. It is difficult to tell the contribution of the monotonic part of the true joint action-value estimator. It has been shown that monotonic functions cannot represent some Q-values.**
>
> As we explained in Section 3, the performance of QTRAN++ depends on the quality of the transformed action-value estimator for approximating the true action-value estimator. The monotonic part helps improve such quality since it implicitly biases the true action-value estimator towards being learned easily by a monotonic function, i.e., the transformed action-value estimator. In our revised paper, we further clarified this point in Section 3.2.
>
> Furthermore, although we agree on the limited representative power of the monotonic part in our true action-value estimator, our true action-value estimator does not suffer from such a limitation since it additionally has a non-monotonic network as a component.
>
> ---
>
> **3. In two out of three scenarios in the ablation studies, FC-QTRAN++ is very similar to QTRAN++.**
>
> We do believe that QTRAN++ demonstrates a concrete improvement over FC-QTRAN++ in our ablation studies; it outperforms FC-QTRAN++ significantly for the last of three scenarios, i.e., 3s_vs_5z. To further alleviate your concern, we refer to our revised paper where we increased the number of scenarios from three to ten. In Figure 4, one can observe a more concrete improvement: QTRAN++ outperforms FC-QTRAN++ for three scenarios, i.e., 10m_vs_11m, MMM2 (negative), 3s_vs_5z, and performs at least as good as FC-QTRAN++ for other seven scenarios.
>
> ---
>
> **4. When training $i$-th head of the transformed action-value estimator, is the utility function of $j$-th agent updated?**
>
> Yes, we update the utility function of $j$-th agent when training $i$-th head of the transformed action-value estimator. In our revised paper, we further clarify this point in Appendix D.
>
> ---
>
> **5. Based on the results from DOP [5] and REFIL [6], I think the multi-head structure may not improve the performance.**
>
> As explained in Section 3.2, our multi-head structure improves the performance of our QTRAN++ by alleviating the issues arising from partial observability. To be specific, it regularizes the agents to rely less on the underlying state information that is not observable during the execution of QTRAN++. While prior works such as DOP and REFIL use multi-head structures similar to ours, their motivations are quite different. DOP aims to reduce the variance in training agent-wise actors using a multi-head critic. REFIL uses a multi-head structure for generalizing the current value factorization to a dynamic number of agents. Hence, their results are inconclusive for assessing the performance improvements from our multi-head structure.

---

### Official Review · AnonReviewer3 · 2020-10-28
**Improvement over QTRAN is demonstrated. Approach is complex. Feels somewhat incremental.**

**Rating:** 6
**Confidence:** 4

**Review:**

### Summary and claims

This work proposes a MARL (multi-agent reinforcement learning) algorithm.
In the MARL setting, multiple agents have to make choices based on independent information to maximize a common objective. An existing algorithm in this space is QTRAN.
The authors propose several modifications to QTRAN: changing the architecture, adding two additional constraints to the loss function and also allowing gradients to flow from the QTRAN objective into the "true" action-value estimator.
The claims of the paper are:
1) QTRAN++ achieves better performance than QTRAN
2) The modifications introduced stabilize training compared to QTRAN

It took me some time to fully understand all of the components that go into QTRAN++ and I find the complexity of the overall algorithm pretty surprising, especially considering that other algorithms in this space are as simple as "add up all the Q values of the individual agents".
I think the proposed changes consist of:
 - Rather than directly training action-value networks, a QMIX-like hypernet approach is used
 - The network that estimates the "true" (combined) action-value is implemented through what the authors call a "semi-monotonic mixing network", which is the sum of a non-monotonic (regular) hypernet and a monotonic hypernet as used in QMIX. This seems pretty arbitrary. Isn't the original idea behind QTRAN that this would accurately track the true values?
 - In QTRAN the separate network that aggregates the Q values of the individual agents is trained to track the "true" action-value. In QTRAN++ this is done through multiple hypernetworks (the authors call these "heads").
 - The loss function is modified to impose two additional constraints on the transforming value function.
 - Gradients are now also backpropagated from the "tracking loss" into the "true" action-value estimator, which makes it somewhat unclear what it is actually representing.

### Relation to prior work

The paper is positioned sufficiently with respect to prior work. I've noticed that there is a larger section on related work in the appendix. I'm not sure what the purpose of moving the related work into the appendix is, especially if some of the papers mentioned there are not actually related to the work presented in this paper. I think it would be good to try to move as much as possible of that section into the main text, leaving out prior work that is not sufficiently related.
Some of the additions in QTRAN++ seem very similar to ideas proposed in QMIX, but this is not directly acknowledged as far as I can tell. It would be good to point out which parts of the architecture come from QMIX.

### Are the claims supported?

The experiments presented in the paper are reasonably thorough and show that
QTRAN++ consistently outperforms QTRAN on the tasks that were tested. SMAC (StarCraft Multiagent Challenge) is a nontrivial benchmark, so I would agree that claim 1) has been shown sufficiently. But I'm not sure whether the ablation studies are thorough enough to really demonstrate that all of the components of the (rather complex) proposed algorithm are really needed.

The authors often make claims about improved stability and other properties of the algorithm throughout the paper, but these are not supported by any empirical evidence. If the authors want to claim that QTRAN++ outperforms QTRAN because of a specific mechanism then it would be good to provide some sort of empirical evidence or proof (the proof in appendix A doesn't count since it doesn't make any statements about stability). Therefore I think that claim 2) is currently not well supported and it would be good to either support it better or soften the statements in the paper to make statements in the form of "we believe that the algorithm has improved stability".

### Presentation and clarity

The paper is reasonably clear and understandable. There are some cases where incorrect grammar or word choice made a sentence difficult to understand. For example the choice of "affluent" to describe a class of estimators. It would be good to address cases like this to improve the clarity of the paper.

### Conclusions

The main claim of the paper (that QTRAN++ is an improvement over QTRAN) has been demonstrated sufficiently. But the high complexity of the approach (with several additions to the algorithm feeling somewhat arbitrary) and the fact that the paper "merely" presents an upgrade to QTRAN could be potential arguments against accepting it.


### *Edit after author comments:*

I have read the author comments and the latest paper revision. The authors have noticeably improved the clarity of the paper in several places and adjusted their claims about the stability of the algorithm, and the improved ablation studies are appreciated. Unfortunately, after thinking it through very carefully and despite the author comments, I have not been able to understand some aspects of the model, for example why gradients from the tracking loss are backpropagated into the value function that is supposed to track the "true" action-values. Several parts of the architecture seem to have a complicated dual purpose, which makes it difficult to understand what is going on and why the model is performing better. I suspect that other readers might also encounter similar issues, which makes it difficult for me to raise my rating. I've decided to leave the rating at 6 (marginal accept).

---

> ### Author Response · Authors · 2020-11-20
> **Response to R3 (2/2)**
>
> **4. I'm not sure whether the ablation studies are thorough enough.**
>
> To alleviate your concerns, we extended our ablation studies from 3 maps to 10 maps. We report the experimental results in Figure 4 of the revised paper. In the figure, one can observe how each component of QTRAN++ provides a solid improvement. Especially, removing any component of QTRAN++ leads to significantly worse performance for at least one of the scenarios. Especially, as mentioned in our previous response, Fix-QTRAN++ underperforms significantly compared to QTRAN++ for all of the considered scenarios.
>
> ---
>
> **5. The authors often make claims about improved stability of the algorithm. It would be good to soften the statements in the paper to make statements in the form of "we believe that the algorithm has improved stability".**
>
> Thank you for pointing this out. We originally described QTRAN++ to improve the “stability” of QTRAN to reflect how our objective provides a denser training signal compared to QTRAN. Nevertheless, we deeply resonate with your concern and revised our paper to soften our claims on improving the stability of the QTRAN++.
>
> ---
>
> **6. There are some cases where incorrect grammar or word choice made a sentence difficult to understand. For example, the choice of "affluent" to describe a class of estimators.**
>
> Thank you for the helpful suggestion. To alleviate your concern, we carefully revised the paper to remove any incorrect grammar or word choice that makes a sentence difficult to understand. For example, we replaced the phrase “more affluent class of estimators” with “a larger class of estimators” in our revised paper.

---

> ### Author Response · Authors · 2020-11-20
> **Response to R3 (1/2)**
>
> We express our deep appreciation for your time and insightful comments. We are grateful for all the positive comments: providing a practical improvement for the theoretically important algorithm (by R2 and R4), strong empirical performance (by you, R1 and R4), novel and general ideas (by R1), and clear writing (by you and R1). In the revised manuscript, we have substantially updated or newly added (Section 2, Section 3, Figure 3, Figure 4, Appendix B, D, E, F) according to the initial reviews and colored them red. In the following, we address your comments one by one.
>
> ---
>
> **1. Using a semi-monotonic mixing network for the true action-value estimator is quite arbitrary. Isn't the original idea behind QTRAN that the true action-value estimator would accurately track the true action-values?**
>
> As we explained in Section 3, the performance of QTRAN++ (and QTRAN) is governed by two factors: (a) quality of true action-value estimator tracking the true action-values and (b) quality of transformed action-value estimator tracking the true action-value estimator. While the original QTRAN focused on designing the true action-value estimator to improve (a), we additionally use the semi-monotonic network to improve (b). To be specific, the semi-monotonic networks improve (b) by implicitly biasing the true action-value estimator to learn a monotonic function that can be easily approximated by the monotonic transformed action-value estimator. In our revised paper, we have further clarified this point in Section 3.2.
>
> We also point out how the semi-monotonic network is necessary for achieving the best performance in our experiments. Indeed, in Figure 4 of our revised paper, one can observe a solid gap between QTRAN++ and FC-QTRAN++ (QTRAN++ without the semi-monotonic network) for three scenarios, i.e., 10m_vs_11m, MMM2 (negative), 3s_vs_5z, and performs at least as good as FC-QTRAN++ for other seven scenarios.
>
> ---
>
> **2. Gradients are now also backpropagated from the "tracking loss" into the "true" action-value estimator, which makes it somewhat unclear what it is actually representing.**
>
> As explained in Section 3, the performance of QTRAN++ depends on the quality of the transformed action-value estimator for tracking the true action-value estimator, i.e., the “tracking loss.” Through the backpropagation, our true action-value estimator can help in minimizing the tracking loss in addition to its original role, i.e., estimating the true action-value.
>
> Furthermore, the effectiveness of the backpropagation is supported by our empirical observation. Indeed, in Figure 4 of our revised paper, one can observe a significant gap between QTRAN++ and QTRAN++ without the backpropagation, i.e., Fix-QTRAN++ for all of the ten scenarios.
>
> ---
>
> **3. I think it would be good to try to move as much as possible of Section B into the main text, leaving out prior work that is not sufficiently related. It would be good to point out which parts of the architecture in QTRAN++ come from QMIX.**
>
> Thank you for the suggestion. To address your concern, we revised the paper accordingly as follows:
> - We moved relevant parts of Section B to Section 2.
> - We left out prior works that are not sufficiently related in Section B.
> - We explicitly stated which parts of our architecture come from QMIX in Section 3.2.

---

### Official Review · AnonReviewer4 · 2020-10-28
**Good paper - simple fixes to an important algorithm, yielding SOTA performance.**

**Rating:** 7
**Confidence:** 3

**Review:**

## Summary

This paper addresses the domain of cooperative multiagent learning with centralised learning and decentralised execution. Specifically, it improves on the QTRAN algorithm, a theoretically justified algorithm which previously had not produced strong learning performance. With these improvements, QTRAN++ outperforms baselines on the SMAC environments.

I recommend accepting this paper. It delivers strong performance on a popular benchmark for complex, cooperative multiagent learning (SMAC). While the algorithmic contribution is incremental, it still delivers insight into how to improve the empirical performance of a theoretically interesting and well-justified algorithm.

 ## Positives

The problem addressed - cooperative multiagent environments with CTDE - is a widely studied and important one. It is well set up in the paper, including discussion of related algorithms.

The base algorithm - QTRAN - should theoretically perform well in a wider variety of environments than other algorithms for these problems, so improving its performance is particularly valuable. The improvements made to the algorithm are clear and well motivated; section 3.1 in particular explains clearly the difference the modified loss is intended to make.

The empirical studies in the paper are strong. They show that QTRAN++ outperforms several baselines in data efficiency and final performance across a variety of domains. Further, a comprehensive ablation study shows that each of the improvements made to QTRAN is independently important (in at least some domains).

## Negatives

The algorithmic contribution of the paper is relatively minor, since it provides fairly simple modifications to an existing algorithm.

Experimentally, it would be good to see experiments on similar domains to those addressed in the original QTRAN paper, which are designed to probe the advantages QTRAN has over related algorithms. This would demonstrate that QTRAN++ retains the benefits of QTRAN in non-monotonic factorisable environments.

## Sources of reviewer uncertainty

I am not knowledgeable enough in this domain to be certain of the coverage of the baselines and domains in the paper. Since the empirical performance of the algorithm is central to the paper, this is important.

---

> ### Author Response · Authors · 2020-11-20
> **Response to R4**
>
> We express our deep appreciation for your time and insightful comments. We are grateful for all the positive comments: providing a practical improvement for the theoretically important algorithm (by you and R2), strong empirical performance (by you, R1 and R3), novel and general ideas (by R1), and clear writing (by R1 and R3). In the revised manuscript, we have substantially updated or newly added (Section 2, Section 3, Figure 3, Figure 4, Appendix B, D, E, F) according to the initial reviews and colored them red. In the following, we address your comments one by one.
>
> ---
>
> **1. The algorithmic contribution of the paper is relatively minor since it provides fairly simple modifications to an existing algorithm.**
>
> Despite being simple, we believe our QTRAN++ to deliver a significant contribution by closing the gap between theory and practice, i.e., QTRAN++ improves the empirical performance of a theoretically interesting algorithm. Furthermore, the algorithmic contribution of QTRAN++ stems not only from proposing the simple modifications but also from effectively combining the modifications to yield a large overall gain in performance. Indeed, as shown in Figure 4, the modifications are complementary, and omitting just one of our modifications results in a degradation of performance for at least one of the considered scenarios. We added the respective discussion in Section 1 of our revised paper.
>
> ---
>
> **2. It would be good to see experiments on similar domains to those addressed in the original QTRAN paper.**
>
> Thank you for the suggestion. To incorporate your comments, we evaluated our QTRAN++ using the multi-domain Gaussian squeeze environment from the original QTRAN paper [1]. The corresponding results are reported in Appendix F of our revised paper. In the experiments, one can observe how QTRAN++ consistently achieves the best result compared to the baselines including QTRAN. Hence, one may conclude that QTRAN++ achieves state-of-the-art performance across different environments and still retains the main strengths of QTRAN.
>
> ---
>
> **References**
>
> [1] Son et al. QTRAN: Learning to factorize with transformation for cooperative multi-agent reinforcement learning, ICML 2019

---

### Official Review · AnonReviewer1 · 2020-10-28
**Interesting work; evaluation can be improved**

**Rating:** 6
**Confidence:** 2

**Review:**

### Summary
This paper presents an improved version of QTRAN [1]. The design is based on new loss function design, as well as new action-value estimator designs. The paper claims superior perfromance gains compared to previous methods on the Starcraft Multi-Agent Challenge (SMAC) environment.

### Strengths
+ The ideas proposed to improve the previous QTRAN (or might be applied to other MARL algorithms as well) seems novel and general.
+ The authors perform comprehensive ablation studies for different components they proposed.
+ The empirical performance on the SMAC benchmark is better and more stable across different runs.
+ The writing is clear and easy to follow.

### Weaknesses
- Only one environment is evaluated, which might not be that convincing. It would be good to see more results on different benchmarks.
- In Figure 3, some results seem to be not converged yet. Since the metric is the win rate, which is bounded, it would be interesting to see given enough training steps, whether all methods can actually converge to similar winning rate, or inherently the proposed scheme can lead to better results.

### Minor issues
Typo: Page 6, "... for being “selfish.” This ..." -> "... for being “selfish”. This ..."


Considering all the aspects, I tend to accept the paper in the current stage.


### Reference
1. Qtran: Learning to factorize with transformation for cooperative multi-agent reinforcement learning. 2019.

----------------------------------------------------------------------

**Updates**: After reading the other reviews and the rebuttal, I still maintain my current score. The additional experiments on the converged results are good to me. As I'm not very familiar with the performance in MARL literatures, I have decreased my confidence from 3 to 2 to reflect some of the concerns of other reviewers involving the performance for the baselines.

---

> ### Author Response · Authors · 2020-11-20
> **Response to R1**
>
> We express our deep appreciation for your time and insightful comments. We are grateful for all the positive comments: providing a practical improvement for the theoretically important algorithm (by R2 and R4), strong empirical performance (by you, R3 and R4), novel and general ideas (by you), and clear writing (by you and R3). In the revised manuscript, we have substantially updated or newly added (Section 2, Section 3, Figure 3, Figure 4, Appendix B, D, E, F) according to the initial reviews and colored them red. In the following, we address your comments one by one.
>
> ---
>
> **1. It would be good to see more results on different benchmarks.**
>
> Thank you for the suggestion. We consider the SMAC benchmark to be sufficient in our experiments since it allows evaluating agents under diverse scenarios. Indeed, the scenarios have varying numbers and types of agents, and require the agents to learn diverse skills such as “kiting” and “focus fire.” For this reason, most prior works such as QMIX [1], and MAVEN [2] only considered the SMAC environment for evaluation.
>
> Nevertheless, to incorporate your comments, we additionally evaluated our QTRAN++ using the multi-domain Gaussian squeeze benchmark [3]. The corresponding results are reported in Appendix F of our revised paper. In the experiments, one can observe how QTRAN++ consistently achieves the best result compared to the baselines. This result is especially significant since the benchmark was designed specifically for demonstrating the strength of the original QTRAN algorithm [3]. This demonstrates that QTRAN++ achieves state-of-the-art performance across different benchmarks and still retains the main strengths of QTRAN.
>
> ---
>
> **2. In Figure 3, some results seem to be not converged yet. It would be interesting to see experiments using enough training steps.**
>
> Thank you for the suggestion. We follow the same number of training steps for the SMAC environment as the prior works, e.g., QMIX [1] and Weighted QMIX [4]. This can avoid a potential reproducibility issue when comparing with the prior works.
>
> Nevertheless, to incorporate your comments, we evaluated our QTRAN++ by increasing the number of training steps twice (two million to four million steps) for the scenarios where algorithms have not converged in Figure 3. The corresponding results are reported in Appendix E of our revised paper. In the experiments, one can observe how our QTRAN++ maintains state-of-the-art performance even after the considered algorithms converge.
>
> ---
>
> **3. Typo: Page 6, "... for being “selfish.” This ..." -> "... for being “selfish”. This ..."**
>
> Thank you for pointing this out. We rephrased the corresponding sentence to alleviate your concern.
>
> ---
>
> **References**
>
> [1] Rashid et al. Monotonic Value Function Factorisation for Deep Multi-Agent Reinforcement Learning, JMLR 2020
> [2] Mahajan et al., MAVEN: Multi-Agent Variational Exploration, NeurIPS 2019
> [3] Son et al. QTRAN: Learning to factorize with transformation for cooperative multi-agent reinforcement learning, ICML 2019
> [4] Rashid et al. Weighted QMIX: Expanding Monotonic Value Function Factorisation, NeurIPS 2020

---

### Author Response · Authors · 2020-11-20
**Common response (updated): our work contributes to bridging the gap between theory and practice of MARL.**

Dear reviewers,

We express our deepest gratitude for your constructive feedback and valuable comments.

As the reviewers highlighted, we strongly believe that our work provides a significant empirical improvement (R1, R3, R4) to a theoretically important algorithm (R2, R4) with novel and general ideas (R1), which is presented with clear writing (R1). Overall, we believe that our work delivers a significant contribution by bridging the gap between theory and practice, for MARL.

In response to the questions and concerns you raised, we have carefully revised and enhanced our paper with the following additional experiments and discussions.

- additional explanation on the contribution of our QTRAN++ for R4 (Section 1)
- softened claims on improving the stability of QTRAN for R3 (Section 1)
- improved description of the prior works for R3 (Section 2 and Appendix B)
- clear explanations for the algorithmic components of QTRAN++ for R2 and R3 (Section 3)
- experiments using QPLEX as an additional baseline for R2 (Figure 3)
- ablation studies on seven more scenarios for R2 and R3 (Figure 4)
- description of the versions of StarCraft and QMIX being used for R2 (Appendix D)
- experiments using an increased number of training steps for R1 (Appendix E)
- experiments on an additional environment for R1 and R2 (Appendix F)
- additional ablation study for the semi-monotonic mixing network on an addition environment for R2 (Appendix G)

The revisions made are marked with “red” in the revised paper.

Thanks, Authors.

---

### Decision · Program_Chairs · 2021-01-07
**Final Decision**

**Decision:**

Reject

**Comment:**

This paper proposes practical improvements to theoretically well founded QTRAN, which is a state-of-the-art technique of cooperative multi-agent reinforcement learning.  The improvements include new designs of loss function and action-value estimator, which might be widely applicable beyond QTRAN.  However, it is not obvious if the proposed improvements actually improves the performance of QTRAN, and experimental evaluation is essential to this work.  After the discussion, there remain some major concerns about the experimental results.  In particular, the performance of baselines in the experiments is not consistent with those reported in the prior work.